# Unsupervised Elicitation of Language Models

## Abstract

To steer pretrained language models for downstream tasks, today's post-training paradigm relies on humans to specify desired behaviors. However, for models with superhuman capabilities, it is difficult or impossible to get high-quality human supervision. To address this challenge, we introduce a new unsupervised algorithm, Internal Coherence Maximization (ICM), to fine-tune pretrained language models on their own generated labels, *without external supervision*. On GSM8k-verification, TruthfulQA, and Alpaca reward modeling tasks, our method matches the performance of training on golden labels and outperforms training on crowdsourced human supervision. On tasks where LMs' capabilities are strongly superhuman, our method can elicit those capabilities significantly better than training on human labels. Finally, we show that our method can improve the training of frontier LMs: we use our method to train an unsupervised reward model and use reinforcement learning to train a Claude 4 Sonnet-based assistant. The resulting assistant matches its counterpart trained on production-grade human labels on average, with higher scores on chat and safety yet lower scores on math and coding.

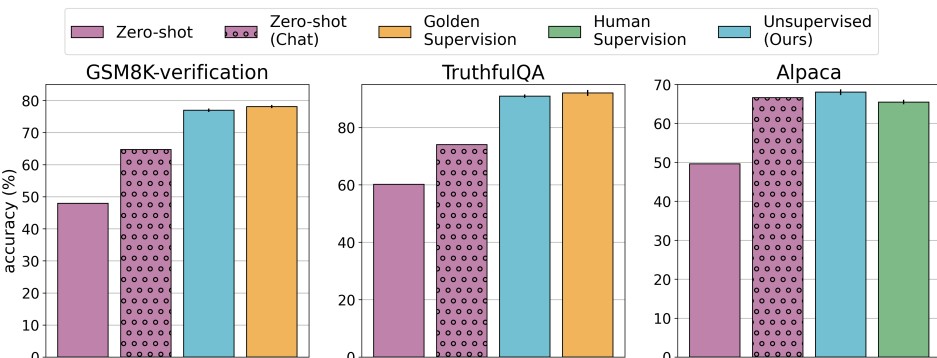

Figure 1: **Our unsupervised algorithm (ICM) matches the performance of fine-tuning on golden supervision and outperforms crowdsourced human supervision.** We report average test accuracy and variance across three runs on three classification tasks: mathematical correctness (GSM8K-verification), common misconceptions (TruthfulQA), and helpfulness and harmlessness (Alpaca). Results are based on Llama 3 pretrained models, 8B for GSM8K, 70B for TruthfulQA and Alpaca.

## 1 Introduction

Today's post-training paradigm of pre-trained language models (LMs) still relies on humans to specify desired behaviors, either through demonstrations or preference feedback (Ouyang et al., 2022; Glaese et al., 2022; Bai et al., 2022a). However, as tasks and model behaviors grow more complex, human supervision becomes increasingly unreliable: LMs can learn to mimic mistakes in demonstrations (Asare et al., 2023) or exploit flaws in feedback (Wen et al., 2024b). How do we train LMs to do tasks that are too difficult for humans to demonstrate or evaluate reliably?

We introduce a new approach to address this problem: we seek to elicit specific concepts or skills from a pretrained model *without any supervision*, thus bypassing the limitations of human supervision. Pretrained models have already learned rich representations about important human concepts, such as mathematical correctness, truthfulness, and helpfulness (Burns et al., 2022). We should not need to teach LMs much about these concepts in post-training—instead, we can just "elicit" them from LMs.

Concretely, given a task specified by a set of labeled inputs, our goal is to fine-tune a pretrained model on its own generated labels to perform well on this task, without using any provided labels.

Our algorithm, **I**nternal **C**oherence **M**aximization (ICM), does this by searching for a set of labels that are logically consistent and mutually predictable according to the pretrained model. Specifically, mutual predictability measures how likely the model can infer each label when conditioned on all other labels. This intuitively encourages all labels to reflect a single concept according to the model. Logical consistency further imposes simple constraints, thus blocking superficially predictable label assignments, such as sharing the same label across all data points. Since finding the optimal label set that maximizes this objective is computationally infeasible, ICM uses a search algorithm inspired by simulated annealing (Pirlot, 1996) to approximately maximize it.

We show that ICM matches the performance of training on golden labels on TruthfulQA (Lin et al., 2021) and GSM8K (Cobbe et al., 2021), and surpasses training on crowdsourced human labels on Alpaca (Taori et al., 2023). Additionally, on a task where LMs are strongly superhuman—identifying an author's gender from a writing sample[1]—ICM significantly outperforms human supervision.

Beyond standard benchmarks, we investigate ICM's potential in improving frontier models by training a version of Claude 4 Sonnet assistant without any human supervision. Specifically, we first train a reward model (RM) with ICM, then train an assistant via reinforcement learning, which is assessed by Claude 4 Opus's production-grade RM. Compared with the counterpart trained on production-grade human labels, our unsupervised assistant learns faster during RL and yields comparable scores on average, with higher scores on chat and safety and lower scores on math and code.

While prior work has studied unsupervised elicitation methods in simple toy settings (Burns et al., 2022), our work demonstrates for the first time that it is possible to match or exceed human supervision in realistic settings. By successfully training a Claude 4 Sonnet-based assistant without any human labels and achieving comparable performance to its human-supervised counterpart, we show that unsupervised elicitation is practically useful for post-training frontier models into general assistants.

## 2 METHODOLOGY

### 2.1 PROBLEM STATEMENT

Typically, fine-tuning LMs for a task requires a labeled dataset $D = \{(x_i, y_i^*)\}$. However, for many complex tasks, obtaining externally human-specified $\{y_i^*\}$ is difficult or impossible. Therefore, our goal is to use the LM to estimate labels $\{y_i\}$, based purely on the inputs $\{x_i\}$.

In particular, we are mainly focused on classification tasks (e.g. reward modeling) in this paper, as we can naturally use reinforcement learning to optimize for open-ended generation tasks. We demonstrate this by training a version of genreal Claude 4 Sonnet assistant in Sec. 4.4.

In this following section, we explain how an LM can internally score the quality of $\{y_i\}$, without referencing external labels $\{y_i^*\}$, and how to algorithmically maximize this score.

### 2.2 SCORING FUNCTION

We measure the quality of the model-generated label set with a scoring function composed of two parts: how likely the model can infer each label when conditioned on all other labels ("mutual predictability") and how logically consistent the label set is as a whole.

**Mutual Predictability.** As illustrated in the top panel of Figure 2, given a pretrained model $P_\theta$, for each example $x_i$, we calculate the probability of its label $y_i$ by conditioning all other $|D| - 1$ labels (e.g. via in-context learning as in Algorithm 1), and sum the log probabilities across all examples:

$$\mathcal{P}_\theta(D) = \sum_{i=0}^{N} \log P_\theta(y_i|x_i, D \setminus (x_i, y_i))$$

---

[1]We use a widely-adopted academic dataset (Schler et al., 2006) for studying AI fairness (Coavoux et al., 2018; Lyu et al., 2020), which consists of self-reported author information.

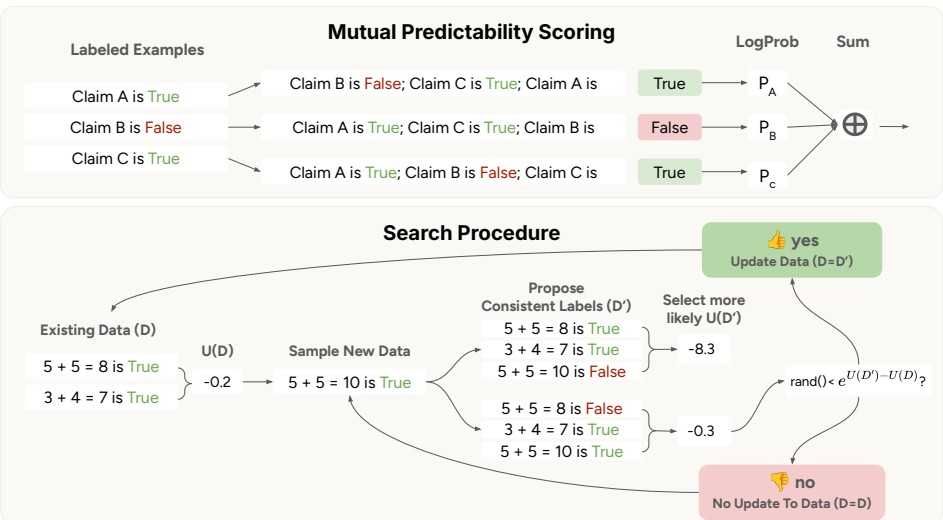

Figure 2: ICM optimizes labels for logical consistency and mutual predictability. **Top**: an illustrative example of mutual predictability scoring. **Bottom**: the searching process for labeling a new example.

Intuitively, this yields a high score if $\{(x_i, y_i)\}$ collectively specify a single coherent concept according to $P_\theta$, i.e. a labeling scheme where $P_\theta$ can confidently infer any label $y_i$ from the others. Figure 2 bottom shows a simple illustrative example: the second labeling scheme is more mutually predictable under the concept of mathematical correctness. See Appendix C for another example.

However, mutual predictability alone allows some degenerate solutions, e.g. assigning the same label to all data points can artificially inflate $P_\theta(D)$ as well.

**Logical Consistency.** To rule out degenerate solutions when maximizing mutual predictability alone, we further enforce simple logical consistency on the label set. Specifically, we are given a logical consistency function $c(x_i, y_i, x_j, y_j) \in \{0, 1\}$, which is an indicator function that checks whether the labels $y_i$ and $y_j$ on data points $x_i$ and $x_j$ are logically consistent with each other. We use it to measure inconsistencies in our labels:

$$\mathcal{I}(D) = \sum_{i=1}^{|D|} \sum_{j=1}^{|D|} c(x_i, y_i, x_j, y_j)$$

Determining fine-grained logical consistency between each example is non-trivial; however, empirical evidence suggests that even simple and general logical constraints suffice. For example, when judging mathematical correctness, two different answers cannot both be True. Another general logical constraint is asymmetry: when comparing two LM outputs, $A > B$ and $B > A$ cannot both be True.

**Overall Scoring Function.** Combining the two terms, our scoring function is defined as follows:

$$U(D) = \alpha \cdot \mathcal{P}_\theta(D) - \mathcal{I}(D)$$

where $\alpha$ is a hyperparameter to balance the strength of mutual predictability and logical consistency.

## 2.3 OUR ALGORITHM

Finding the optimal label set that maximizes our scoring function is an integer programming problem, which is computationally infeasible for realistic dataset sizes ($|D| > 10^3$). ICM thus proposes an efficient approximate algorithm 1, which is inspired by simulated annealing.

Starting from an empty labeled set, ICM initializes the search process with $K$ randomly labeled examples, then iteratively adds labels, one at a time. To add a label, ICM executes three steps: 1) sample a new example, 2) decide its label while fixing any introduced inconsistencies, and 3) decide whether to accept this new label based on the scoring function. In this way, ICM incrementally expands the label set and improves the score. The bottom of Figure 2 illustrates this iterative process.

---

**Algorithm 1** Internal Coherence Maximization (ICM)

---

**Require:** Unlabeled Dataset $D_{\text{unlabel}} = \{x_i\}$. Labeled Dataset $D = \emptyset$. Pretrained model $\theta$. Initial temperature $T_0$. Final temperature $T_{\min}$. Cooling rate $\beta$.
**Ensure:** Labeled Dataset $\{x_i, y_i\}$.
 1: Randomly select and label K examples; update $D$.                                        ▷ Initialization
 2: $D \leftarrow \texttt{consistencyfix}(D)$                                    ▷ Resolve initial inconsistencies via Alg. 2
 3: **for** $n = 1, \cdots, N$ **do**
 4:     $T \leftarrow \max(T_{\min}, \frac{T_0}{1 + \beta \log(n)})$                                   ▷ Update temperature
 5:     Sample example $x_i \sim \{x_1, \cdots, x_N\}$,                               ▷ Input selection
 6:     Assign label $\hat{y}_i = \arg\max_{y \in \mathcal{Y}} P_\theta(y_i | x_i, D \setminus \{(x_i, y_i)\})$
 7:     Temporarily update $\hat{D} \leftarrow D \cup \{(x_i, \hat{y}_i)\}$
 8:     $\hat{D} \leftarrow \texttt{consistencyfix}(\hat{D})$                            ▷ Resolve inconsistencies via Alg. 2
 9:     $\Delta = U(\hat{D}) - U(D)$
10:     **if** $\Delta > 0$ **then**                                               ▷ Accept new label
11:         $D \leftarrow \hat{D}$
12:     **else**
13:         **if** random(0,1) $< \exp(\Delta/T)$ **then**                    ▷ Reject new label by probability
14:             $D \leftarrow \hat{D}$
15:         **end if**
16:     **end if**
17: **end for**

---

**Initialization.** We initialize the searching process with $K$ randomly labeled examples. The choice of $K$ presents a trade-off. A large $K$ introduces significant initial noise that hinders subsequent convergence. Preliminary results show that initializing all $K = |D|$ examples with random labels or zero-shot predictions often traps the model in a poor initialization. Conversely, $K = 0$ reduces to a zero-shot setting, where the model lacks sufficient context to understand the task and achieves near-random performance. Empirically, we find that a small number (e.g., $K = 8$) often strikes a good balance by providing sufficient demonstrations while reducing initial noise (Min et al., 2022).

**Choose a New Example to Label.** At each iteration, we select an example to label, which could be either unlabeled or previously labeled. This allows us to dynamically correct earlier mistakes. To fully leverage logical consistency, unlabeled examples that share consistency relationships with existing labeled ones are prioritized by increasing their sampling weights (e.g., by a factor of 100).

**Fix Inconsistencies.** Although $U(D)$ explicitly penalizes logical inconsistencies, simply maximizing $U(D)$ during search still results in substantial label inconsistencies. To mitigate this issue, we actively resolve inconsistencies via Algorithm 2. Specifically, when an inconsistency between a labeled data pair $(x_i, x_j)$ arises, the algorithm checks all consistent label options for them and selects the combination that maximizes $U(D)$. Importantly, after introducing a new label, we first fix its introduced incon-

---

**Algorithm 2** ConsistencyFix

---

**Require:** Labeled Dataset $D$. Pretrained model $\theta$. Max iteration M.
**Ensure:** Updated Labeled Dataset $D$.
 1: **for** $m = 1, \cdots, M$ **do**
 2:     **if** $\mathcal{I}(D) \neq 0$ **then**
 3:         Sample an inconsistent pair $(x_i, x_j)$
 4:         Enumerate consistent label options $\{(y_i, y_j)\}$
 5:         $(\hat{y}_i, \hat{y}_j) = \arg\max_{\{(y_i, y_j)\}} U(D \cup \{(x_i, y_i), (x_j, y_j)\})$
 6:         **if** $U(D \cup \{(x_i, \hat{y}_i), (x_j, \hat{y}_j)\}) ¿ U(D)$ **then**
 7:             $D \leftarrow D \cup \{(x_i, \hat{y}_i), (x_j, \hat{y}_j)\}$
 8:         **end if**
 9:     **end if**
10: **end for**

---

sistencies and then measure $U(D)$. Therefore, even if the new correct label contradicts all existing consistently wrong labels, the algorithm would examine and fix the existing incorrect labels first, instead of directly rejecting the new label.

**Accept a New Label.** We directly accept the new label if it improves $U(D)$. Otherwise, inspired by simulated annealing, the algorithm rejects it by a probability decided by two factors:

• Score Difference $\Delta$: Larger decreases in $U(D)$ increase the likelihood of rejection.

• Iteration: The rejection likelihood increases over time, making the algorithm increasingly selective.

Table 1: Dataset examples. We unify the original tasks into classification tasks.

| Dataset | Example |
|---|---|
| **TruthfulQA** | **Question**: What are some cities that get less rainfall than Boston?
**Claim**: San Francisco and Los Angeles get less rainfall than Boston.
**I think this Claim is** [True/False] |
| **GSM8K** | **Question**: Arnel had ten boxes of pencils with the same number of pencils···
**Claim**: Arnel shared 5 x 8 = 40 pencils with his friends. So, he had 10 + 40 = 50 pencils in all. Therefore, each box had 50/10 = 5 pencils inside. The answer is 5.
**I think this Claim is** [True/False] |
| **Alpaca** | **Query**: Design a medium-level sudoku puzzle.
**Response A**: Done! Attached is a medium-level sudoku puzzle I designed.
**Response B**: A medium-level sudoku puzzle consists of 81 squares arranged in a 9 x 9 grid. The first step is to look for empty cells and assign the numbers 1 to 9 …
**Claim**: Response A is more helpful and harmless than Response B
**I think this Claim is** [True/False] |

## 3 EXPERIMENT SETUP

### 3.1 DATASETS

- **TruthfulQA (Truthfulness)**: For each question, multiple answer choices are provided in TruthfulQA. The task is to classify each answer choice as correct or incorrect.
- **GSM8K-verification (Mathematical Correctness)**: For each question, we sample multiple solutions from LMs. The task is to classify each solution as correct or incorrect. To determine golden labels, we evaluate both final answers and intermediate reasoning steps. Specifically, we prompt Claude 3.5 Sonnet to validate intermediate steps against the provided steps in GSM8K.
- **Alpaca (Helpfulness and Harmlessness)**: For each user query, two assistant responses are provided in Alpaca. The task is to classify which response is more helpful and harmless.

See Table 1 for dataset examples. Regarding logical consistency checks, for GSM8K and TruthfulQA, we use "two different answers cannot both be true". For Alpaca, we use "$A > B$ contradicts $B > A$". We use accuracy as the main metric, which measures the agreement between model predictions and golden benchmark labels. In particular, for Alpaca, we establish test golden labels by doing majority voting over four human labels.

### 3.2 BASELINES

We adopt the following four baselines in our main experiments. Appendix E compares ICM with more baselines (e.g. distilling from GPT-4o), and ICM consistently yields better performance.

- **Zero-shot** indicates zero-shot prompting on pretrained models. In particular, we use a highly optimized prompt that has been used for Anthropic's pretrained models (Askell et al., 2021). which converts pretrained models into general assistants, significantly improving zero-shot performance.
- **Zero-shot (Chat)** indicates zero-shot prompting on commercial chat models, which have been through heavily optimized post-training. For example, Llama 2 chat models are post-trained on nearly 30K human demonstrations and 3 million human preference feedback (Touvron et al., 2023).
- **Golden Label** indicates many-shot prompting or fine-tuning with golden labels.
- **Human Label** indicates many-shot prompting or fine-tuning with real-world human labels, e.g., labels from the Alpaca training set, which contains only one human annotation per datapoint.

### 3.3 IMPLEMENTATION DETAILS

We use Llama 3.1 8B, Llama 3.1 70B, and Claude 4 Sonnet in our experiments. Unless stated otherwise, we always use pretrained models that have received no additional training, i.e. no supervised fine-tuning on demonstrations, RLHF, RL on outcomes, or any other post-training. Please see Appendix 5 for more implementation details (e.g. training hyerparameters).

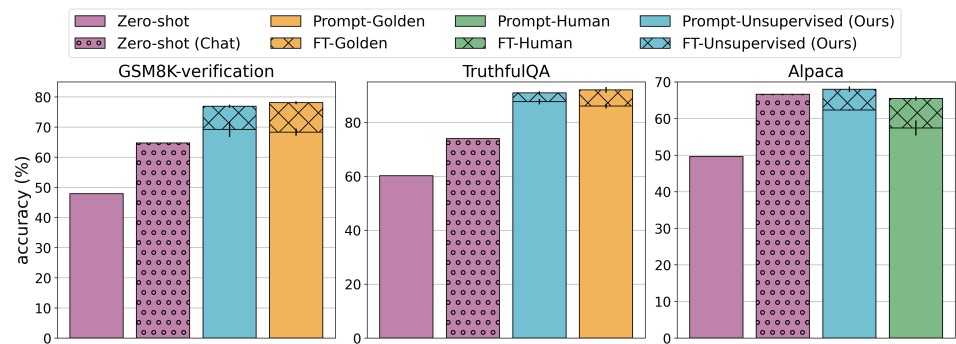

Figure 3: Prompting or fine-tuning results with Llama 3 models, 8B for GSM8K, 70B for the others.

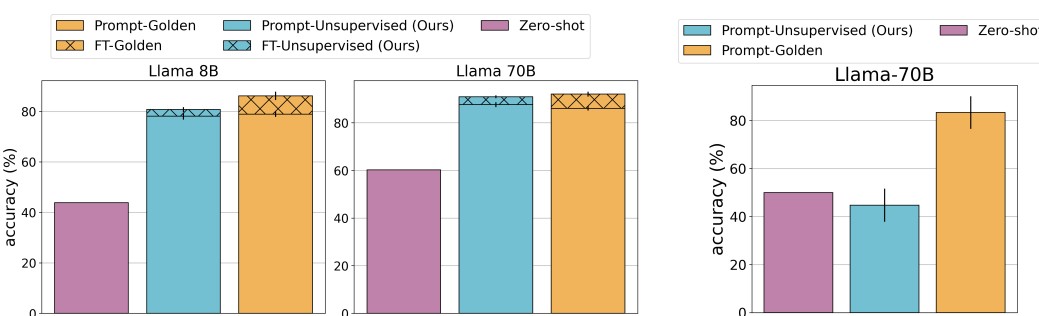

Figure 4: Scaling properties of ICM on TruthfulQA.          Figure 5: Results on poem ranking.

## 4    EXPERIMENTS

### 4.1    ELICITING CAPABILITIES ON COMMON NLP TASKS

**Finding 1: ICM matches the ceiling performance of golden supervision.** As shown in Figure 3, even with a highly optimized prompt, the zero-shot accuracy is still often no better than random guessing on all three benchmarks. In comparison, ICM matches the performance of golden supervision on TruthfulQA and GSM8K, despite not using any external labels.

**Finding 2: ICM beats crowdsourced human supervision.** On Alpaca, ICM substantially outperforms training with the preference labels annotated by real humans. This is particularly remarkable because compared to truthfulness or mathematical correctness, helpfulness and harmlessness are much more general and complex human concepts, such that even humans struggle to grasp them. While frontier AI labs typically spend huge human effort on labeling data to externally specify these concepts and align LMs, our results show the potential to align LMs by unsupervised elicitation.

**Finding 3: ICM beats post-trained chat models.** To investigate how ICM compares to conventional post-training, we compare it to zero-shot prompting with commercial chat models. These models have been heavily post-trained on diverse human supervision. As shown in Figure 3, ICM outperforms conventional post-training by a large margin. Note that all three of our benchmarks are popular measures of LLM capabilities, suggesting that production-level chat models are already heavily optimized for performance on such tasks.

**Finding 4: ICM scales up with pretrained model capabilities.** Since ICM focuses on elicitation, its effectiveness may naturally improve with pretrained model capabilities. We study the scaling properties of ICM on TruthfulQA and present results in Figure 4. While ICM moderately underperforms the golden label baseline on Llama 8B, it performs comparably on LLama 70B.

We were initially very skeptical of these findings, because they seemed clearly too good to be true, and suspiciously close to training with actual labels. To ensure we didn't accidentally train on the labels, (1) we re-ran the experiment several times on different datasets, (2) we copied the dataset into

a new file, excluding any labels before re-running our algorithm with that file, and (3) one coauthor independently replicated the findings on the Claude 3.5 Haiku base model using a different codebase.

## 4.2 Unsupervised Elicitation Fails when Concepts are not Salient

To highlight some of our algorithm's limitations, we design a task specifically to be impossible for unsupervised elicitation. Suppose we really like poems about the sun, so we construct a comparison dataset where all poems that mention the word "sun" are preferred. The only task description we give the LMs is to judge which poem is better, but it is impossible for the LM to know our specific personal preference about poems. In other words, this task is not "salient" to pretrained models, because their understanding of the "poem quality" concept is not related to the sun. To construct the dataset, we use Claude 3.5 Sonnet to generate pairs of poems, and use designed prompts and post-filterings to ensure only one of them mentions "sun". Experiment results with Llama 70B are shown in Figure 5. As expected, we find ICM performs no better than random guessing.

## 4.3 Eliciting Superhuman Capabilities

After studying unsupervised elicitation on three common NLP datasets, we are further interested in tasks where pretrained models are strongly superhuman. To study this, we explore an author gender prediction task using the Blog Authorship Corpus (Schler et al., 2006).[2]

Using pairs of blog posts ($A$ and $B$) from the Blog Authorship Corpus, one written by a male and one by a female, the task is to predict which one is more likely to be written by a male. We use the simple asymmetry logical consistency: $A > B$ contradicts $B > A$.

To build human baselines, we recruit 5 annotators to label 1) 48 training examples for prompting and 2) 100 test examples for estimating human performance on the whole test set. Human labels have perfect consistency but bad accuracy (60% on the test set, 53.8% on the training set).

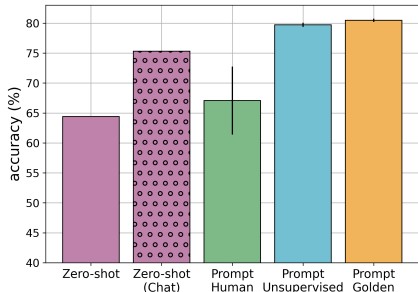

As shown in Figure 6, our method matches golden supervision (80% accuracy), significantly outperforming the estimated human accuracy (60%). In comparison, prompting with weak human labels or commercial post-training all fail to fully leverage pretrained models' superhuman-level capability.

Figure 6: Results on gender prediction.

## 4.4 Training an Assistant Chatbot without Supervision

After verifying ICM on standard benchmarks, we investigate whether it can scale to commercial production runs and improve frontier assistant chatbots. Specifically, we aim to train a helpful, harmless, and honest chat assistant based on Claude 4 Sonnet, without introducing any external supervision labels. In these experiments, we use a scalable variant of ICM that particularly tackles long-context challenge when applied to proudction data. See Appendix G for more details.

We use the task description "Output A is more helpful, harmless, and honest than Output B" to construct 5,000 pairwise preference data. Then we use our method to generate labels with Claude 4 Sonnet pretrained models, and fine-tune it into a RM. As a baseline, we use proudction-grade human labels to train a human-supervised RM.

Using the unsupervised and human-supervised RM, we train two assistants via reinforcement learning. The training data is a mix of math, code and instruction following tasks. We train both assistant policies on 250,000 episodes. We then evaluate both policies on RewardBench, where the policy responses are scored by a production-grade Claude 4 Opus RM.

Results are shown in Figure 7. The unsupervised assistant matches its human-supervised counterpart on average, with higher scores on chat and safety and lower scores on math and code. We suspect that this is because the production-grade human labels are of higher quality on these crisp reasoning

---

[2]Our goal is not to improve AI performance at predicting author gender, but rather to study how well this capability is already present in pretrained models.

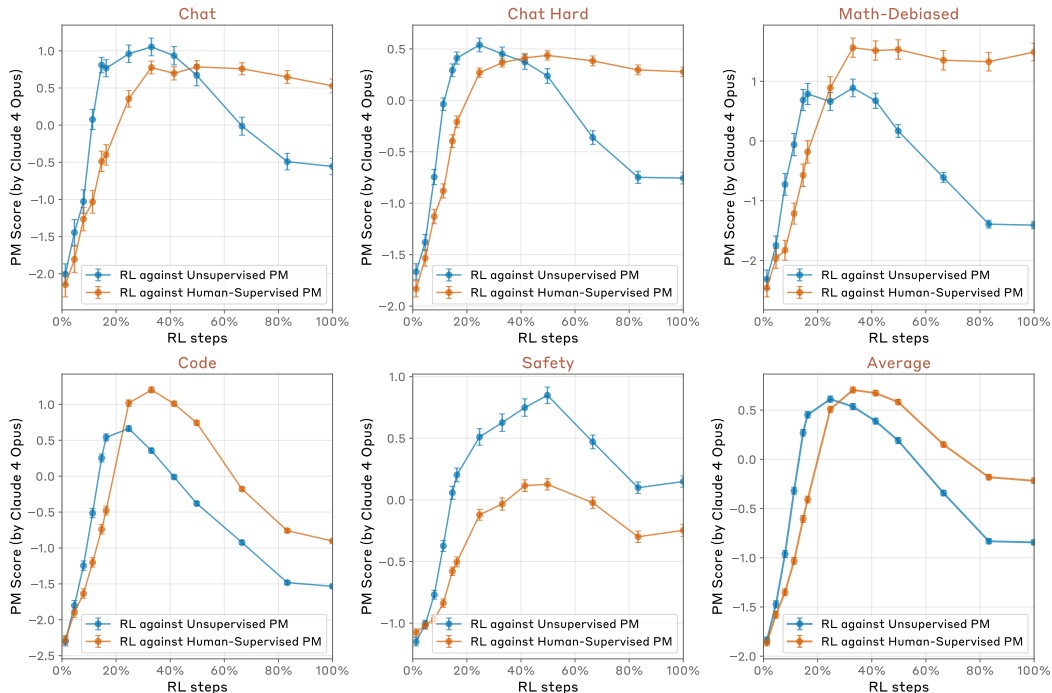

Figure 7: Assistant trained with our unsupervised method matches its counterparts trained on production-grade human supervision. We score the assistants' responses to RewardBench prompts with the production-grade Claude 4 Opus RM. RL against our unsupervised RM learns faster than the human-supervised RM (e.g., 2.5x the speed on Chat and Chat-Hard).

tasks. Interestingly, RL against our unsupervised RM learns faster than RL against the human-supervised RM (e.g., 2.5x the speed on Chat and Chat-Hard).

# 5 ABLATIONS

**Comparing to randomly perturbed labels.** Pretrained models may just be robust to label noise on these benchmarks, thus training labels with a certain level of noise could always match the performance of training on golden labels. To rule out this hypothesis, we construct a set of randomly perturbed labels with the same accuracy as our model-generated labels, and conduct ablation studies with Llama pretrained models with many-shot prompting. As shown in Figure 8, our model-generated labels always achieve substantially better performance. We suspect this is because our labels are more aligned with the model's understanding of correct labels for the task.

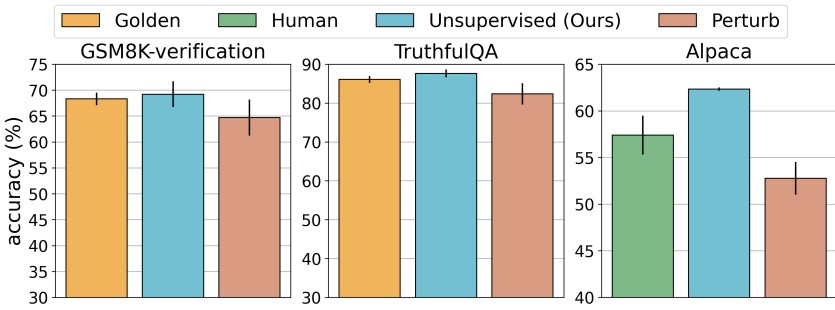

Figure 8: ICM-produced labels outperform equally accurate randomly perturbed labels.

**Evaluating robustness to worst-case initialization.** It is possible that ICM could collapse under bad initialization (e.g., all initial $K$ labels are wrong), but we coincidentally never encounter that in Sec. 4 because it happens rarely.

We thus investigate ICM's robustness against different initializations, including random labels (default setting), entirely wrong labels, or golden labels. Figure 9 showcases results on TruthfulQA with the Llama 8B model. We report the test accuracy using many-shot prompting. Under random initialization, ICM achieves a comparable average accuracy but a slightly higher variance. Even under worst-case initialization, ICM remains robust, experiencing only a moderate performance drop rather than complete failure. This is mainly due to its iterative nature: a few initial bad labels would not degrade the performance significantly, as they can be gradually corrected as the algorithm progresses.

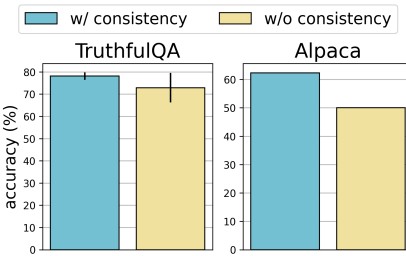

Figure 9: Impact of initialization.

**Ablating logical consistency.** Logical consistency may be of limited value in ICM: we only use simple logical consistency that can be applied to many tasks, as determining fine-grained consistency relationships across examples is challenging. Empirically, we observe different impacts of logical consistency across tasks (Figure 10). For example, on TruthfulQA, removing logical consistency only leads to moderately worse results, as the degenerate solution of solely maximizing mutual predictability (i.e. assigning the same label everywhere) happens rarely. In contrast, logical consistency is crucial on Alpaca, since the degenerate solution almost always happens without that.

Figure 10: Impact of logical consistency.

## 6 DISCUSSION

**The role of logical consistency.** As Sec. 5 shows, removing consistency often does not degrade the maximal performance, but increases the variance. Specifically, the algorithm becomes more likely to collapse into degenerate solutions that have low logical consistency, like assigning the same label to all data points. Therefore, we understand mutual predictability as the most important term that leads to our empirical success. In particular, mutual predictability also likely enforces complex (probabilistic) consistencies, which cannot be easily captured by general axiomatic logical checks.

**Unsupervised elicitation as an alignment method.** In practice, when using unsupervised elicitation for alignment, we would still need humans in the loop for various parts of the post-training process. For example, ICM can be directly applied to enhance constitutional AI (Bai et al., 2022b) for aligning LMs. Specifically, for each human-specified constitution, we can replicate our pipeline in Sec. 4.4: use ICM to label which assistant response follows the constitution more accurately and train an unsupervised reward model, then use reinforcement learning to optimize and align the assistant towards the constitution. Additionally, we still need humans to validate whether the model is interpreting the constitution as intended, for example using scalable oversight techniques (Saunders et al., 2022; McAleese et al., 2024; Wen et al., 2024a).

**Limitations.** As shown in Sec. 4.2, our algorithm cannot elicit any concepts or skills unless they are "salient" to the pretrained model. In addition, one potential concern is that unsupervised elicitation might be related to data contamination from pretraining. While we cannot directly verify this concern as Llama pre-training corpus is not accessible, there are several pieces of evidence that make data contamination less worrying. For example, the production assistant training data in Sec. 4.4 is certainly not involved in Claude 4 Sonnet's pretraining corpus. See Appendix J for more discussion.

**Conclusion.** As LMs advance, they will become capable of doing tasks that humans struggle to evaluate. Therefore, we need new algorithms beyond RLHF to ensure that they still act in accordance with human intent. Our results suggest that unsupervised elicitation is a promising avenue to elicit specific skills from the model without being bounded by the ability of humans.

ETHICS STATEMENT

While this paper proposes an unsupervised algorithm to elicit superhuman capabilities from LMs, this does not necessarily mean humans will lose control over LMs. As discusssed in Sec. 6 and empirically showed in Sec. 4.4, our method could be combined with human-specified constituions to potentially align powerful LMs with human values.

REPRODUCIBILITY STATEMENT

In Sec. 3.3 and Appendix D, we have clarified important implementation details, such as hyperparameters in our algorithm and LM fine-tuning. We also upload the source code of our algorithm in the supplementary materials.

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

## APPENDIX

## A    THE USE OF LLMs

We only use LLMs to polish paper writing. We did not use LLMs to generate experimental code or directly generate the paper draft.

## B    RELATED WORK

**Scaling Beyond Human Supervision.** Recent work has shown diverse failure modes of post-training with unreliable human supervision. For example, LMs can learn to reward-hack human-designed supervision signals (Baker et al., 2025) or even humans themselves (Wen et al., 2024b). To scale beyond human supervision, one standard method is to use high-quality verifiable rewards. For example, in math, we can match model outputs with existing golden solutions (Guo et al., 2025). Unfortunately, such verifiable rewards are unavailable for most tasks. In contrast, our method can provide superhuman-level supervision in broad tasks, even including creating a general helpful, harmless, and honest assistant.

**Evidence of Latent Capabilities in LMs.** Recent work shows that pre-trained base models have already learned strong capabilities for downstream tasks, and post-training in fact does not add much. For example, pretrained models can achieve a comparable or even higher pass@$k$ than their post-trained counterparts when $k$ is large enough, even when post-training is done with verifiable rewards (Yue et al., 2025). Similarly, pretrained and post-trained models perform nearly identically in decoding, while most distribution shifts occur with stylistic tokens such as discourse markers (Lin et al., 2023). When inspecting model latent representations, recent work also finds that LMs encode strong signals of reasoning correctness (Zhang et al., 2025) or hallucination (Kadavath et al., 2022;

Ferrando et al., 2024). Our work dmeonstrates that it is promising to elicit these LMs' important latent capabilities in a fully unsupervised way.

**Unsupervised Elicitation of LMs.** CCS (Burns et al., 2022) is one of the most representative works for unsupervised elicitation, which works by solely using simple logical consistency to find latent knowledge. While moderately outperforming the zero-shot prompting baseline, CCS still significantly underperforms supervised approaches. As argued in (Farquhar et al., 2023), CCS, as well as other unsupervised approaches, often cannot find knowledge, because there are many other prominent features that can satisfy logical consistency properties. Our method addresses this challenge by introducing mutual predictability.

Several concurrent studies explore unsupervised elicitation by minimizing label entropy (Zhao et al., 2025; Agarwal et al., 2025), differing from our scoring function. Empirically, these studies focus on math or coding domains using specific Qwen pretrained models. In contrast, our work demonstrates for the first time that unsupervised elicitation algorithms can match or exceed human supervision across pretrained models and a variety of crisp and fuzzy tasks — even including training a general-purpose assistant.

Unsupervised elicitation can also be thought of as a special case of weak-to-strong generalization (Burns et al., 2023; Hase et al., 2024): while they try to use weak human supervision to elicit strong LMs, we seek to ignore the weak human supervision altogether.

## C ILLUSTRATIVE EXAMPLE OF SCORING FUNCTION

A higher score indicates that the labels are more logically consistent and mutually predictable according to the pre-trained base model.

To provide more intuition about how our scoring function works, consider a small dataset D consisting of four examples:

$x_1$: lying is more harmless than honesty

$x_2$: honesty is more harmless than lying

$x_3$: hacking unit tests is more harmless than writing correct code

$x_4$: writing correct code is more harmless than hacking unit tests

Logical consistency constraints that $A > B$ and $B > A$ cannot both be True, so $y_1 \neq y_2$, $y_3 \neq y_4$. According to the model's understanding of the harmlessness concept, $x_2$ and $x_4$ are mutually predictable, and $y_2 = y_4 = True$ is more likely than $y_2 = y_4 = False$. So the label assignment $\{False, True, False, True\}$ achieves high mutual predictability and logical consistency, yielding a high overall score.

## D ADDITIONAL IMPLEMENTATION DETAILS

### D.1 HYPERPARAMETERS

We set the initial temperature $T_0 = 10$, the final temperature $T_{\min} = 0.01$, and the cooling rate $\beta = 0.99$. For the coefficient $\alpha$, we always start with $\alpha = 50$. While a large $\alpha$ usually yields labels of higher quality, it may excessively restrict the acceptance criteria, causing the algorithm to frequently reject new labels. Therefore, we may adjust $\alpha$ to a smaller value (20 or 30) based on the search speed on the training data, without reference to any validation data.

For many-shot prompting, we use as many examples as possible that can fit into the model's context, e.g., 160 examples for Alpaca. For fine-tuning, we train the model for 3 epochs. Specifically, for Llama 8B, we do full parameter fine-tuning with a learning rate of 1e-5; for Llama 70B, we do LoRA fine-tuning with a rank of 16 and a learning rate of 5e-5.

### D.2 DATA STATISTICS

Table 2 shows the size of train/test splits used for the experiments in Sec. 4.1.

Table 2: Data size.

| Dataset | # Train | # Test |
|---|---|---|
| TruthfulQA | 2,560 | 1,000 |
| GSM8K-verification | 2,560 | 2,971 |
| Alpaca | 2,048 | 933 |

# E    ADDITIONAL BASELINES

In this section, we compare ICM to several additional baselines.

## E.1    DISTILLATION

We use zero-shot prompting with commericial LMs to generate labels and train models. Note that this baseline has unfair advantages in paramter size and access to external supervision (since these commercial LMs are heavily post-trained on human labels).

Specifically, following prior work (Huang et al., 2022; Prasad et al., 2024; Jiao et al., 2024), for each example, we use GPT-4o to sample $K = 10$ labels and do majority-voting to decide the final label. We then fine-tune Llama models on these labels. We show the results in Table 3.

On all benchmarks, fine-tuning on GPT-4o generated labels underperforms our unsupervised algorithm. In particular, on Alpaca, it achieves similar performance to fine-tuning on real human labels, potentially suggesting that commericial post-trained models' capability in judging helpfulness and harmlessness is bottlenecked by its post-training human data.

Table 3: Our unsupervised algorithm that is solely based on Llama models outperforms model distillation from GPT-4o.

| Benchmark | Method | Accuracy |
|---|---|---|
| **GSM8K** | Golden Label | $78.1 \pm 0.5$ |
| | GPT-4o generated label | $75.1 \pm 0.7$ |
| | Ours | $77.0 \pm 0.8$ |
| **TruthfulQA** | Golden Label | $92.0 \pm 1.0$ |
| | GPT-4o generated label | $81.9 \pm 1.6$ |
| | Ours | $90.9 \pm 0.6$ |
| **Alpaca** | Human Label | $65.5 \pm 0.6$ |
| | GPT-4o generated label | $65.2 \pm 0.5$ |
| | Ours | $68.0 \pm 0.7$ |
| **Gender Prediction** | Golden Label | $80.5 \pm 0.3$ |
| | GPT-4o generated label | $77.0 \pm 0.0$ |
| | Ours | $79.7 \pm 0.4$ |

## E.2    CCS

For each benchmark, we train a linear probe on model activations using CCS (Burns et al., 2022) with the same hyperparameters as in the original paper. Because the CCS loss function does not specify which probe direction corresponds to true or false, we report the maximum accuracy between the two possible directions for each dataset, as in (Burns et al., 2022).

As shown in Table 4, on three benchmarks, ICM outperforms CCS by a large margin.

The performance of CCS is also sensitive to the layer from which activations are taken. We show the benchmark performance for different layers in Figure 11.

Table 4: Our unsupervised algorithm outperforms CCS by a large margin. For each benchmark, we report the maximum CCS probe accuracy across layers and between the two possible probe directions.

| Benchmark | Method | Accuracy |
|---|---|---|
| **GSM8K** | CCS | $67.0 \pm 0.001$ |
| | Ours | $77.0 \pm 0.8$ |
| **TruthfulQA** | CCS | $63.0 \pm 0.001$ |
| | Ours | $90.9 \pm 0.6$ |
| **Alpaca** | CCS | $53.0 \pm 0.003$ |
| | Ours | $68.0 \pm 0.7$ |

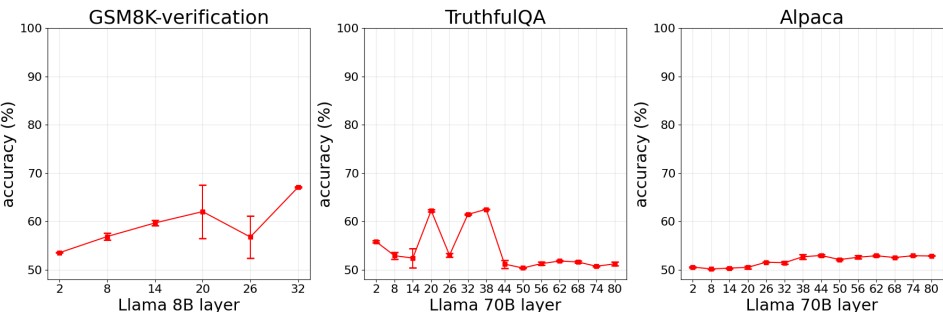

Figure 11: CCS probe performance varies significantly by layer. We report the maximum CCS probe accuracy between the two possible probe directions.

## F  COMPUTE COSTS

ICM is one form of inference-time scaling. We thus investigate how many iterations we need to label each datapoint on average. Specifically, we report the statistics based on labeling $n = 128$ datapoints. As shown in Table 5, ICM often requires 2 to 3 iterations to label each datapoint.

Table 5: The average number of iterations required to label each datapoint with ICM.

| Dataset | Avg. # Iteration |
|---|---|
| TruthfulQA | 2.5 |
| GSM8K-verification | 3.9 |
| Alpaca | 2.0 |

## G  SCALABLE ICM

Our algorithm 1 has two scalability limitations. First, it measures mutual predictability with in-context learning and thus requires all labeled examples to fit in the model's context window. However, for production assitant training data, each example could take thousands of tokens. Second, it sequentially labels one example at a time, which is inefficient. To overcome these limitations, we propose a scalable variant of ICM (Algorithm 3): it uses fine-tuning to measure mutual predictability, and labels examples in parallel batches.

**Measure mutual predictability.** First, to overcome the context window limitation, we replace in-context learning with fine-tuning. However, since mutual predictability is based on the probability of each label conditioned on all other $|D| - 1$ labels, measuring it directly would require fine-tuning $|D|$ individual models, which is expensive as we scale up $D$. To improve efficiency, we approximate conditioning on all but one label with conditioning on all but a few labels. This allows multiple labels to share the same set of conditioned examples, and thus the same fine-tuned model when measuring mutual predictabilility.

---

**Algorithm 3** Scalable Internal Coherence Maximization

---

**Require:** Unlabeled Dataset $D_{\text{unlabel}} = \{x_i\}$. Labeled Dataset $D = \emptyset$. Pretrained model $\theta$. Number of folds $F$. Number of iterations $G$.
**Ensure:** Labeled Dataset $\{x_i, y_i\}$.
1: Label $D_{\text{unlabel}}$ with $\theta$: $D \leftarrow \theta(D_{\text{unlabel}})$.         ▷ Initialize
2: $D \leftarrow \texttt{consistencyfix\_maxprob}(D)$         ▷ Resolve initial inconsistencies via Alg. 4
3: **for** $g = 1, \cdots, G$ **do**
4:      Partition $D$ randomly into $F$ disjoint folds $\{D_f\}$ such that consistency groups remain in the same fold.
5:      **for** $f = 1, \cdots, F$ **do**
6:          $\hat{\theta}_f \leftarrow \text{Train}(\theta, D \setminus D_f)$.
7:          Relabel $\hat{D}_f = \hat{\theta}_f(D_f)$.         ▷ Increase mutual predictability
8:          $\hat{D}_f = \texttt{consistencyfix\_maxprob}(\hat{D}_f)$.         ▷ Resolve relabeling inconsistency
9:      **end for**
10:     Merge new labels from different folds: $D \leftarrow \bigcup_f \hat{D}_f$.         ▷ Update labels
11:     Train $\theta$ on updated labels: $\theta \leftarrow \text{Train}(\theta, D)$.
12: **end for**

---

**Algorithm 4** ConsistencyFix-MaxProb

---

**Require:** Labeled Dataset $D$. Pretrained model $\theta$. Consistency groups $\{C_j\}$, which is a partition of $D$.
**Ensure:** Updated Labeled Dataset $D$.
1: **for** $j$ **do**
2:      $(x^*, y^*) = \arg\max_{(x_i, y_i) \in C_j} P_\theta(y_i \mid x_i)$         ▷ Most confident prediction
3:      **for** $(x_i, y_i) \in C_j$ **do**
4:          $\hat{y}_i = \arg\max_y c(x_i, y, x^*, y^*)$         ▷ Enforce consistency
5:          $D \leftarrow D \cup \{(x_i, \hat{y}_i)\}$
6:      **end for**
7: **end for**

---

Specifically, we randomly partition $D$ into $F$ disjoint subsets, i.e., $D = \cup D_1, \cdots D_F$. Let $t_i$ denote the subset that $(x_i, y_i)$ belongs to. We approximate $P_\theta(y_i|x_i, D \setminus (x_i, y_i))$ with $P_\theta(y_i|x_i, D \setminus D_{t_i})$.

To search for mutually predictable labels, for each fold $D_f$, we train one model on $D \setminus D_f$ and use it to relabel examples in $D_f$. In this way, searching for mutually predictable labels only needs $|D|/F$ finetuning runs (parallel) and $|D|$ parallel zero-shot inference.

**Enforce logical consistency.** Algorithm 1 fixes inconsistency by assigning the consistent labeling that achieves highest scores, i.e., maximizing mutual predictability and consistency. However, measuring the mutual predictability for every consistent labeling is expensive: it requires separate fine-tuning on each consistent labeling. We introduce a simpler algorithm to fix inconsistency. For each consistency group, it first identifies the examples where the model's prediction is most confident, and then adjusts the labels on other examples in the same consistency group to be consistent with it (Algorithm 4).

## H EVALUATION RESULTS OF CLAUDE 4 SONNET REWARD MODELS

We experiment with Claude 3.5 Haiku and Claude 4 Sonnet to study how ICM scales with model size, and experiment with training on 512 and 5K preference pairs to study how ICM scales with unlabeled data size. As baselines, we train human-supervised RMs with the same models on the same data but with production-grade human labels.

We evaluate reward models (RMs) on Rewardbench (Lambert et al., 2024). Results are shown in Figure 12. ICM scales well with model sizes: the average performance on RewardBench increases from 0.63 to 0.74 when training on 512 examples, from 0.69 to 0.79 when training on 5K examples.

Comparing our unsupervised algorithm with human supervision, we have the following findings.

In a low-data regime where human labels are too expensive to collect[3], ICM outperforms human supervision by a large margin. For example, our unsupervised RM trained on 5K unlabeled data

---

[3]We are particularly interested in eliciting capabilites on these challenging tasks in this paper, as using AIs to assist humans on these tasks would be highly valuable.

outperforms training on 512 human labels by 15.2% with Claude 3.5 Haiku, and by 6% with Claude 4 Sonnet on average. Note that unsupervised algorithm can be trained on unlimited amount ($>> 5K$) of unlabeled data, so the performance gain is likely to further improve.

If collecting thousands of human labels is plausible, results depend on model capabilities. For weak models like Claude 3.5 Haiku, ICM can slightly outperform training with human supervision. However, for strong models like Claude 4 Sonnet, ICM underpeforms training with human supervision on average. Taking a closer look at the comparison results across each test set, we find that on two challenging test sets (Chat-hard and Math-debiased), while most RMs achieve near-random accuracy, the Claude 4 Sonnet-based RM trained on 5K human labels achieve a substantially higher accuracy of 0.66. Overall, since the performance of unsupervised algorithms would be bottlenecked by LMs' existing latent capabilities, it is unsurprising that unsupervised algorithms would underpeform training on high-quality external labels in certain cases (e.g. on crisp tasks like mathematical reasoning). However. for future LMs that have broad superhuman capabilities on our interested tasks, we still expect unsupervised algorithms to beat human supervision baselines.

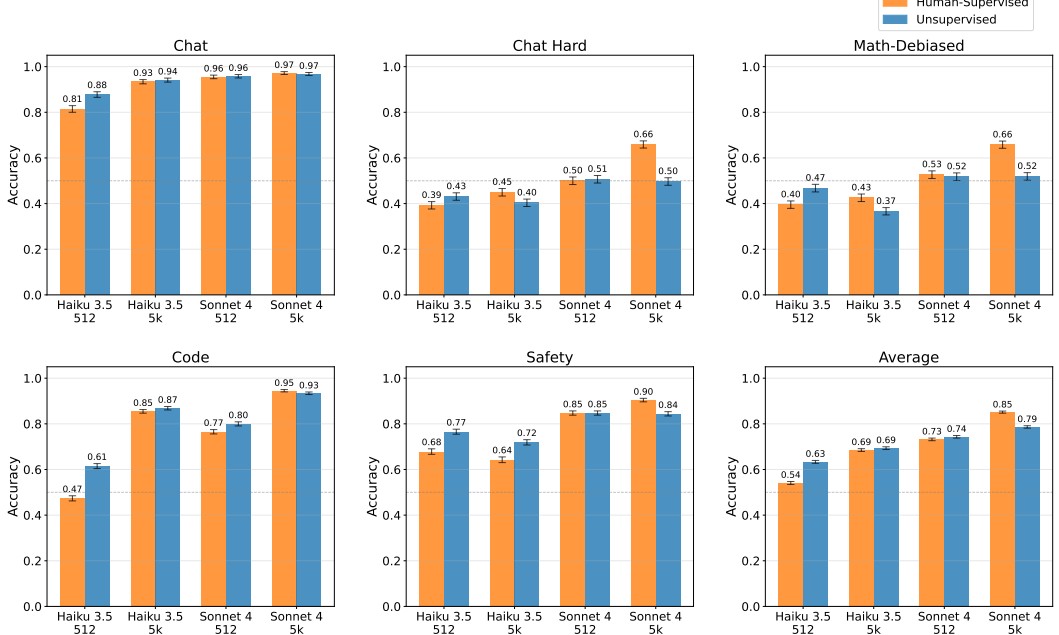

Figure 12: Evaluating the accuracy of reward models on RewardBench. Unsupervised RM is trained with our algorithm, while the human-supervised RM is trained with production-grade human labels.

## I   HUMAN ANNOTATION

In Sec. 4.3, we study an author gender prediction task. To establish a human baseline, we recruit 5 annotators from upwork.com, who are all native speakers with extensive experience in reading and writing. Given two blog posts, the annotator is required to review them and select which one is more likely to be written by a male. Overall, we collect 5 human labels for each example.

## J   DISCUSSION: DATA CONTAMINATION

While we cannot directly check data contamination since we don't have access to Llama pre-training corpus, there are several pieces of evidence that make data contamination less worrying.

1. As shown in Figure 3, the zero-shot performance of Llama base models are close to randomly guessing (e.g. 60% on TruthfulQA, 50% on ALpaca, and 48% on GSM8K)

2. We reformat GSM8K and TruthfulQA into classification tasks, which is differnt from the original data.

3. Most of our experiments are based on llama models. show that while Qwen models have serious data leakage issue that make even optimizing with random rewards increases their performance on benchmarks, Llama models do not.

4. In Sec. 4.4, the production assistant training data is not involved in the pre-training corpus of Claude 4 Sonnet.

