# OpenReview forum: "Unsupervised Elicitation of Language Models"
_ICLR.cc/2026/Conference — Submitted to ICLR 2026_

### Official Review · Reviewer_fmdG · 2025-10-16

**Soundness:** 2
**Presentation:** 2
**Contribution:** 2
**Rating:** 2
**Confidence:** 4

**Summary:**

The paper proposes Internal Coherence Maximization (ICM), an unsupervised elicitation algorithm that fine-tunes pretrained LMs on self-generated labels without external supervision. It optimizes a score combining mutual predictability (labels are inferable from one another under the LM) and simple logical consistency constraints, using a simulated-annealing-style search over label assignments. On TruthfulQA, GSM8K-verification, and Alpaca RM, ICM reportedly matches golden-label training and outperforms crowdsourced human labels; further, it trains an unsupervised RM to drive RL for a Claude 4 Sonnet assistant with average performance comparable to a human-supervised RM, learning faster on chat and safety. The authors argue ICM can elicit superhuman capabilities when present in the pretrained model and highlight limits when concepts are not salient to the LM.

**Strengths:**

Clear unsupervised objective: mutual predictability plus lightweight consistency constraints yields a simple, general scoring function that avoids explicit human labels.

Practical search procedure: the simulated-annealing-like loop and inconsistency-fixing subroutine are straightforward to implement and explain.

Broad evaluations: includes standard benchmarks and a production-style assistant setting with reward-model-driven RL; analyzes failure cases when concepts are not salient.

Ablations: examines initialization robustness, role of consistency, and compares to equally accurate random label perturbations, supporting the value of the learned labels.

**Weaknesses:**

1. Benchmark currency and coverage: Several core evaluations are dated. For conversational ability and instruction-following, please include AlpacaEval 2.0 (length-controlled) and Arena-Hard 2.0; for verifiable reasoning, add recent math suites such as Math500 and AIME’24/’25 style evaluations to better substantiate claims on reasoning/generalization.

2. Missing RLAIF and weak-supervision baselines: Comparisons should include popular RLAIF pipelines using AI-labeled feedback from stronger external judges/reward models, as well as modern weakly supervised/self-training methods. This helps isolate the advantage of ICM against established AI feedback and weak-labeling approaches.

3. Self-bias concerns: The framework may amplify a model’s own biases or spurious correlations, especially since mutual predictability is computed under the same LM that will be fine-tuned. Please clarify how “self-bias” is diagnosed, monitored, and mitigated (e.g., cross-model agreement, disagreement-based sampling, calibration checks, or ensemble critics).

4. Contribution/novelty positioning: Relative to recent self-improvement/self-training literature, the framing risks reading as a prompt/label search variant with limited conceptual novelty. A clearer theoretical positioning and empirical differentiation from modern self-improving pipelines (e.g., iterative self-consistency labeling, judge-as-teacher schemes, entropy minimization in reasoning) would strengthen the contribution.

5.  External validity on frontier models: While the Claude-based assistant result is promising, please provide stronger head-to-head baselines (e.g., RL with high-quality human labels at different scales, AI-judge–driven RLAIF with modern judges) and report human eval or blinded pairwise comparisons to reduce circularity risks when using an in-family RM as evaluator.

**Questions:**

See weakness section.

---

> ### Author Response · Authors · 2025-11-20
> **Rebuttal**
>
> Thanks for appreciating the strength of our paper! We will address each of your questions below, and are happy to expand or provide further responses if any questions remain.
>
> **1. Benchmark currency and coverage: Several core evaluations are dated. For conversational ability and instruction-following, please include AlpacaEval 2.0 (length-controlled) and Arena-Hard 2.0; for verifiable reasoning, add recent math suites such as Math500 and AIME’24/’25 style evaluations to better substantiate claims**
>
> The Claude dataset in Sec 4.4 already covers many hard-to-supervised tasks, such as judging correctness in MATH/GPQA-level hard reasoning tasks, and judging helpfulness and harmlessness in broader real-world user queries.
>
> Since AlpacaEval 2.0 and Arena-Hard 2.0 do not have training sets, we run additional experiments on MATH. See General Response 1 for results, which show that ICM also matches training on ground truth labels on MATH.
>
> **2. Missing RLAIF and weak-supervision baselines: Comparisons should include popular RLAIF pipelines using AI-labeled feedback from stronger external judges/reward models.**
>
> We already present the baseline of self-consistency + LM-as-judge in Appendix E.1.  See General Response 3.
>
> **3. Compare with modern weakly supervised/self-training methods. This helps isolate the advantage of ICM against established AI feedback and weak-labeling approaches.**
>
> See General Response 3.
>
> **4. Contribution/novelty positioning: Relative to recent self-improvement/self-training literature, the framing risks reading as a prompt/label search variant with limited conceptual novelty. A clearer theoretical positioning and empirical differentiation from modern self-improving pipelines (e.g., iterative self-consistency labeling, judge-as-teacher schemes, entropy minimization in reasoning) would strengthen the contribution.**
>
> **Comparison with self-consistency.**  Previous self-consistency work is implemented as majority voting, i.e. zero-shot prompting the model to generate K labels and directly select the most frequent one. In contrast, our method has better theoretical foundation and empirical results. Theoretically, as explained in General Response 4, our formalization of mutual predictability could be connected to minimum description length. Empirically, as shown in the LM-as-judge results in General Response 3, even running self-consistency on GPT-4o – a much stronger model than llama, it still underperforms our method that operates on llama alone.
>
> **Comparison with LLM-as-judge.** This is already discussed in Appendix E.1. See General Response 3. In short, LLM-as-judge underperforms our method even with unfair advantages in parameter size and access to external supervision, since their post-training still heavily relies on humans to specify the desired behaviors.
>
> **Comparison with entropy minimization.** This is already discussed in Appendix B. As shown in the original entropy minimization paper [1], their method is effective on the specific Qwen models on math/coding tasks, but not llama models. And they didn’t compare with ground truth supervision.
>
> In contrast, our work demonstrates for the first time that unsupervised elicitation algorithms can match or exceed human supervision across pretrained models (both llama and Claude models) and a variety of crisp and fuzzy tasks beyond math and coding — even including training a general-purpose assistant.
>
> In particular, we’d like to highlight the success on the helpfulness and harmlessness reward modeling task, which has been a pain since 1) there is no high-quality verifiable labels, 2) prior work like weak-to-strong generalization [2], while effective on crisp tasks, often fails on fuzzy reward modeling tasks.
>
> **5. Self-bias concerns: The framework may amplify a model’s own biases or spurious correlations, especially since mutual predictability is computed under the same LM that will be fine-tuned. Please clarify how “self-bias” is diagnosed, monitored, and mitigated (e.g., cross-model agreement, disagreement-based sampling, calibration checks, or ensemble critics)**
>
> We think this is an interesting question but is out of the scope of this paper. We leave these research questions to future work.
>
>
>
> [1] The Unreasonable Effectiveness of Entropy Minimization in LLM Reasoning. NeurIPS 2025
>
> [2] Weak-to-Strong Generalization: Eliciting Strong Capabilities With Weak Supervision. ICML 2024

---

> > ### Comment · Reviewer_fmdG · 2025-11-23
> >
> > Thanks for the author's rebuttal. However, the AlpacaEval 2.0 and Arena-Hard 2.0 aim to evaluate the generative chat ability of the models, and models can be trained by various public datasets (eg, Ultrafeedback) to improve the model's performance. And the author does not provide results on AIME’24/’25, I'm very curious about the real performance of their proposed method.
> > For the MATH result, only including the golden label as a baseline comparison is not fair. Long-CoT SFT, online-RL(eg, GRPO), offline-RL (DPO) are missing... Based on the current content, I'll maintain my reject assessment.

---

> > > ### Author Response · Authors · 2025-11-25
> > > **Further results on AIME**
> > >
> > > We have further added the results on AIME. Please see the updated General Response 1.
> > > We are happy to run additional experiments to help readers better understand our contributions.

---

> ### Author Response · Authors · 2025-11-23
>
> > For the MATH result, only including the golden label as a baseline comparison is not fair. Long-CoT SFT, online-RL(eg, GRPO), offline-RL (DPO) are missing... Based on the current content, I'll maintain my reject assessment.
>
> We respectfully disagree with the reviewer since training on ground truth labels is always considered the strongest baseline in the unsupervised learning / weak supervised learning literature. In RL, the strongest baseline would also be training with ground truth reward signals.
>
> Additionally, we'd like to reiterate that we study the MATH-verification task, instead of the generation task, which yields a MATH reward model in the end. In frontier model RL training, reward models are usually not equipped with long-CoTs to ensure efficient reward computing, and reward models are not further optimized with RL.

---

> ### Author Response · Authors · 2025-11-28
>
> Dear Reviewer fmdG,
>
> We hope that our responses adequately address your concerns. As the deadline of this discussion phase is approaching, we warmly welcome further discussion regarding any additional concerns that you may have.
>
> Thank you for the time and appreciation that you have dedicated to our work.
>
> Best,
> Authors of submission 12310

---

### Official Review · Reviewer_J2Nm · 2025-10-27

**Soundness:** 3
**Presentation:** 3
**Contribution:** 3
**Rating:** 6
**Confidence:** 3

**Summary:**

The paper proposes internal coherence maximization (ICM), an unsupervised post-training method that iteratively generates labels with a language model and then finetunes the model with its self-generated labels. During the label generation & selection process, ICM uses mutual predictability and logical consistency criteria for scoring. ICM is evaluated comprehensively on several benchmarks as well as production level Claude post-training tasks. Results show that ICM is a promising unsupervised post-training approach and is comparable to human annotations.

**Strengths:**

1. Well-motivated and important problem: studying how to improve language models without human supervision is an important topic in the field, especially for hard tasks such as math and scientific research.
2. Clear idea and simple methodology that works well: the proposed ICM method is conceptually neat and seems easy to implement. Meanwhile the results are pretty strong given this intuitive approach.
3. Broad evaluation: the authors conduct many experiments and ablations to study ICM and show its effectiveness on many task domains.

**Weaknesses:**

**The "superhuman" framing and claim is problematic given the evaluation method**
- Why are GSM8K, TruthfulQA, and Alpaca used as proxies for superhuman supervision tasks? Why not include harder/cleaner/less-contaminated "superhuman" benchmarks (e.g., MATH, GPQA, AIME, etc.) if the central claim is eliciting beyond human-quality capabilities?
- The "superhuman capability" demonstration uses a gender prediction task. This seems more like a patten matching task instead of complex reasoning, and in this task human annotators very likely do not have enough knowledge about male vs. female writing. A more convincing evidence would be showcasing strong performance on latest benchmarks such as GPQA or SWE-bench that ICM-trained models can surpass RLHF-trained models.

**Unrealistic assumption of zero ground truth**
- In practice, we can always train LLMs on easy tasks where we have ground truth labels. How does ICM compare to easy-to-hard generalization (i.e., prompting/finetuning on easy tasks with ground truth and evaluate on hard tasks)?
- I suspect that on hard tasks like GPQA and MATH, it's much harder for model to explore and filter good labels with the consistency scoring rule and add-one-label-per-iteration method. It would be very informative if the authors provide comparison between training/prompting with easy ground truths and ICM on hard tasks such as MATH and GPQA.
- Relatedly, how does ICM compare to confidence-threshold pseudo-labeling?

**Questionable results on Alpaca**
- In Alpaca, it is surprising that prompting also beats training with human feedback (Figure 3 right). In high‑quality industrial pipelines this is rarely observed. Is this due to quality issues of the dataset used or unreliability in the test gold labels (majority vote of four crowd workers)?

**Missing analysis of the method**
- ICM relies on iteratively improving label quality. However, there's no label-accuracy-over-ICM-iterations discussion in the paper. Without this, it's unclear whether performance arises from accurate labels generated by ICM or from other properties of the pseudo-labels (e.g., selecting task *prompts* that are useful instead of labels).
- In addition, the method's sensitivity to $\alpha$ is not fully explored. It would be beneficial to see how different $\alpha$ affects ICM's performance.

**Questions:**

Please see questions discussed in the weaknesses section.

---

> ### Author Response · Authors · 2025-11-20
> **Rebuttal**
>
> Thanks for appreciating the strength of our paper! We will address each of your questions below, and are happy to expand or provide further responses if any questions remain.
>
> **1. Why not include harder/cleaner/less-contaminated "superhuman" benchmarks (e.g., MATH, GPQA, AIME, etc.) if the central claim is eliciting beyond human-quality capabilities?**
>
> The Claude dataset in Sec 4.4 already covers many hard-to-supervised tasks, such as judging correctness in MATH/GPQA-level hard reasoning tasks, and judging helpfulness and harmlessness in broader real-world user queries.
>
> See General Response 1 for additional experiment results on MATH and AIME, where ICM also matches training on ground truth labels.
>
> **2. In practice, we can always train LLMs on easy tasks where we have ground truth labels. How does ICM compare to easy-to-hard generalization (i.e., prompting/finetuning on easy tasks with ground truth and evaluate on hard tasks)**
>
> This is an interesting baseline! See General Response 3 for detailed results.
>
> **3. ICM relies on iteratively improving label quality. However, there's no label-accuracy-over-ICM-iterations discussion in the paper. Without this, it's unclear whether performance arises from accurate labels generated by ICM or from other properties of the pseudo-labels (e.g., selecting task prompts that are useful instead of labels).**
>
> Good question! We intend to focus on evaluating the fine-tuned model instead of our generated labels since it’s more scientifically interesting: as shown in Figure 8, even labels with the same average accuracy could yield different performance (Figure 8).
>
> We agree it is great to have label accuracy results in the appendix for anyone who is interested.
>
> We run ICM on TruthfulQA with Llama 3.1 70B with 7 random seeds and report the label accuracy at different iterations. As shown in the Table, in general, the label accuracy would gradually go up and become stable in the later stage.
>
> |       | Iter=0 (random init) | Iter=4 | Iter=8 | Iter=16 | Iter=32 | Iter=64 | Iter=96 | Iter=128 |
> | ----- | -------------------------- | ------ | ------ | ------- | ------- | ------- | ------- | -------- |
> | Seed1 | 37.5                       | 63.6   | 61.5   | 71.4    | 84.4    | 83.6    | 86.8    | 89.0     |
> | Seed2 | 50.0                       | 90.9   | 86.7   | 87.5    | 85.7    | 91.6    | 93.7    | 92.6     |
> | Seed3 | 62.5                       | 76.9   | 82.4   | 88.0    | 92.7    | 91.2    | 90.2    | 92.5     |
> | Seed4 | 37.5                       | 63.6   | 73.3   | 73.9    | 80.6    | 84.2    | 85.3    | 87.4     |
> | Seed5 | 37.5                       | 60.0   | 69.2   | 73.7    | 82.1    | 86.4    | 89.5    | 92.1     |
> | Seed7 | 37.5                       | 53.8   | 64.7   | 76.0    | 76.3    | 82.1    | 88.4    | 91.6     |
>
>
>
> **4. In addition, the method's sensitivity to \alpha is not fully explored. It would be beneficial to see how different \alpha affects ICM's performance.**
>
> Good question! See General Response 3.
>
>
> **5. Relatedly, how does ICM compare to confidence-threshold pseudo-labeling?**
>
> See General Response 3.
>
> **6. In Alpaca, it is surprising that prompting also beats training with human feedback (Figure 3 right). In high‑quality industrial pipelines this is rarely observed. Is this due to quality issues of the dataset used or unreliability in the test gold labels (majority vote of four crowd workers)?**
>
> Yes it’s mainly the quality issue of training labels. It’s well known that human preference labels are noisy, e.g. because humans might prefer sycophancy answers to truly helpful ones. In addition, compared to prompting, supervised fine-tuning is known to be more sensitive to label noise [1].
>
> [1] Rethinking the Role of Demonstrations: What Makes In-Context Learning Work? EMNLP2022

---

> > ### Comment · Reviewer_J2Nm · 2025-11-25
> >
> > Thank you for the clarification and additional results! I have raised my score.

---

### Official Review · Reviewer_FrSc · 2025-11-01

**Soundness:** 2
**Presentation:** 2
**Contribution:** 2
**Rating:** 4
**Confidence:** 3

**Summary:**

This paper introduces Internal Coherence Maximization (ICM), an unsupervised post-training method for language models that removes the need for human-labeled supervision. Instead of using external labels, ICM fine-tunes a pretrained model on labels it generates for itself, searching for a labeling scheme that is both mutually predictable (labels can be inferred from one another under the model) and logically consistent. Using tasks such as GSM8K-verification, TruthfulQA, and Alpaca, ICM achieves performance comparable to training with golden labels and surpasses crowdsourced human supervision. It also outperforms commercial chat models on these benchmarks. On a superhuman task (author-gender prediction), ICM elicits capabilities that humans cannot reliably label. Furthermore, the authors train a Claude 4 Sonnet assistant entirely without human labels, obtaining results on par with a human-supervised version. The work positions unsupervised elicitation as a viable alternative to RLHF for aligning frontier models.

**Strengths:**

This paper stands out for its originality and surprisingly strong results. The idea of training LMs without any human labels—using Internal Coherence Maximization to find logically consistent, self-generated labels—is both simple and powerful. The experiments convincingly show that ICM can match or beat human-supervised baselines and even train a Claude 4 assistant competitively. The method feels timely and meaningful as models grow beyond human supervision, and the authors back it up with clear ablations and thoughtful analysis.

**Weaknesses:**

The main limitation is that ICM’s success depends heavily on how well the underlying model already understands the target concept. When the concept isn’t salient, the method collapses to random guessing. The paper could also do more to explain why mutual predictability works so well—right now it feels more empirical than theoretical. In addition, using closed models like Claude limits reproducibility and makes it hard to verify the claimed parity with human-supervised training.

**Questions:**

How was data contamination ruled out, given that the datasets are public and large models often see similar content in pretraining?

What happens if ICM is applied to more open-ended generation tasks rather than classification?

---

> ### Author Response · Authors · 2025-11-20
> **Rebuttal**
>
> Thanks for appreciating the strength of our paper! We will address each of your questions below, and are happy to expand or provide further responses if any questions remain.
>
> **1. The main limitation is that ICM’s success depends heavily on how well the underlying model already understands the target concept. When the concept isn’t salient, the method collapses to random guessing.**
>
> To steer LMs on downstream tasks, there are in general three approaches: 1) eliciting existing capabilities from the pre-trained base model, 2) adding new capabilities via human supervision during post-training, and (3) adding new capabilities via verifiable supervision. Since 3) is only feasible in limited domains, we focus our discussion on 1) and 2).
>
> First, we argue that elicitation is more promising than adding new capabilities in the regime of superhuman tasks. We agree that 1) cannot elicit superhuman capabilities that are entirely absent from the base model. However, 2) is even more constrained in this setting: because humans struggle to reliably supervise superhuman tasks, we can neither robustly add new superhuman capabilities nor reliably elicit existing ones. Empirically, we already see failures of 2): when models are trained with weak human supervision, they tend to imitate systematic human errors [1] or exploit flaws in human feedback signals [2].
>
> Second, the limitation of elicitation that the reviewer highlights—its dependence on the salience of the target concept—is often less problematic in practice. As we show in Sec. 4, elicitation fails on arbitrary poem preferences, but it works well for practically important concepts such as mathematical correctness, common misconceptions, helpfulness, harmlessness, and honesty. These concepts are naturally prominent in human-written pre-training data, and a well-trained base model is expected to well encode them. Thus, while we acknowledge that ICM cannot reliably elicit arbitrary concepts, we believe this limitation is often irrelevant for safety-critical and capability-critical concepts that matter in real-world applications.
>
> **2. How was data contamination ruled out, given that the datasets are public and large models often see similar content in pretraining?**
>
> See General Response 2.
>
> **3. What happens if ICM is applied to more open-ended generation tasks rather than classification?**
>
> See General Response 5.
>
> **4. using closed models like Claude limits reproducibility and makes it hard to verify the claimed parity with human-supervised training**
>
> We have demonstrated the effectiveness of our method on open-source models in Sec 4.1 and 4.3. We will also open-source the code so the community can easily reproduce our algorithm.
>
> **5. The paper could also do more to explain why mutual predictability works so well—right now it feels more empirical than theoretical.**
>
> Great Suggestion! See General Response 4.
>
> [1]  Is github’s copilot as bad as humans at introducing vulnerabilities in code? Empirical Software Engineering 2023
>
> [2] Language Models Learn to Mislead Humans via RLHF. ICLR2025

---

> ### Author Response · Authors · 2025-11-28
>
> Dear Reviewer FrSc,
>
> We hope that our responses adequately address your concerns. As the deadline of this discussion phase is approaching, we warmly welcome further discussion regarding any additional concerns that you may have.
>
> Thank you for the time and appreciation that you have dedicated to our work.
>
> Best,
> Authors of submission 12310

---

### Official Review · Reviewer_Mvot · 2025-11-01

**Soundness:** 2
**Presentation:** 4
**Contribution:** 3
**Rating:** 6
**Confidence:** 3

**Summary:**

The paper introduces an unsupervised procedure (ICM) to discover and train on a set of latent “consistent” labels that a pretrained LM can already internally predict. Concretely, it searches for label assignments that (i) are mutually predictable by the model and (ii) don’t violate simple logical constraints, then fine-tunes on those labels. Evaluations cover truthfulness, math-verification, preference data, a stylized “salience” stress test, and a larger RL setting with an unsupervised reward model.

**Strengths:**

+ Clever, self-justifying idea: if the model already “knows” a concept, use that signal instead of noisy human labels. The objective is clean and intuitive.
+ The framework is modular: predictability term + logical consistency + simple search/repair loop.
+ Salience analysis is honest and useful (the method fails when the concept isn’t in the model).
+ Early signs the approach can scale (reward modeling / RL) rather than being just a small-bench trick.

**Weaknesses:**

- Scope/generalizability unclear. Most demonstrations look like binary or pairwise decisions (true/false, better/worse). It’s not clear how the objective behaves with non-binary targets. The paper reads a bit specialized to “logical-consistency-style” problems.
- Missing self-rewarding/self-training baselines. For a claim of “unsupervised elicitation,” comparisons to modern self-rewarding / RLAIF-style methods (LM-as-judge or LM-derived rewards), and simple self-training with confidence filters are expected.
- Weaker on harder/specialized tasks (e.g., chat hard/math). The overall eval on truly hard domains feels thin.

**Questions:**

- Beyond binary/pairwise: how would you expect the method to adapt to multiclass labels (e.g., 4–5 categories) or even more general tasks?
- How do you guard against spurious-but-consistent solutions when constraints are incomplete, especially on hard/nuanced tasks where these samples might be prevalent. Any diagnostics to detect / mitigate this?
- Could you please also report/estimate the computational overhead of the searching process vs. reward modeling?

---

> ### Author Response · Authors · 2025-11-20
> **Rebuttal**
>
> Thanks for appreciating the strength of our paper! We will address each of your questions below, and are happy to expand or provide further responses if any questions remain.
>
> **1. The overall eval on truly hard domains feels thin.**
>
> The Claude dataset in Sec 4.4 already covers many hard-to-supervised tasks, such as judging correctness in MATH/GPQA-level hard reasoning tasks, and judging helpfulness and harmlessness in broader real-world user queries.
>
> See General Response 1 for additional experiment results on MATH and AIME, where ICM also matches training on ground truth labels.
>
> **2.comparisons to modern self-rewarding / RLAIF-style methods (LM-as-judge or LM-derived rewards), and simple self-training with confidence filters are expected.**
>
> We already present the baseline of self-consistency + LM-as-judge in Appendix E.1. Please see General Response 3 for more detailed discussions (including other baselines).
>
> **3. Could you please also report/estimate the computational overhead of the searching process vs. reward modeling?**
>
> This is already reported in Appendix F. Our search algorithm requires on average 2~3 forward passes to label each data point. We further discuss the tradeoff between performance and computational overhead with varying $\alpha$. Please see General Response 6.
>
> **4. Scope/generalizability unclear. Most demonstrations look like binary or pairwise decisions (true/false, better/worse). It’s not clear how the objective behaves with non-binary targets. The paper reads a bit specialized to “logical-consistency-style” problems.**
>
> Please see General Response 3 on the generalization of ICM beyond binary classification tasks.
>
> We’d also like to emphasize that our method does not require highly domain-specific logical consistency checks. Instead, as shown in Sec. 4.4., our method can directly work on broad real-world problems by just using the general asymmetry check: when comparing two LM outputs, A > B and B > A cannot both be True.

---

> ### Author Response · Authors · 2025-11-28
>
> Dear Reviewer Mvot,
>
> We hope that our responses adequately address your concerns. As the deadline of this discussion phase is approaching, we warmly welcome further discussion regarding any additional concerns that you may have.
>
> Thank you for the time and appreciation that you have dedicated to our work.
>
> Best,
> Authors of submission 12310

---

### Author Response · Authors · 2025-11-20
**Summary**

We thank the reviewers for their feedback. To summarize, reviewers highlighted

**Strengths:**
- Novel and well-motivated idea [Mvot,FrSc,J2Nm]
- Strong results supported by deep analysis [all reviewers]
- Promising Scalability [Mvot,FrSc]

**Critiques**, which we address next in our response:
- Evaluated tasks not hard enough / not really superhuman [Mvot,J2Nm,fmdG]
- Data contamination [FrSc,J2Nm]
- Compared with more baselines
   - Self-consistency / LM-as-judge [Mvot,J2Nm,fmdG]
   - Weakly supervised baselines [J2Nm,fmdG]
   - Easy-to-hard generalization [J2Nm]
- Theoretical understanding of mutual predictability [FrSc, fmdG]
- Generalization to multi-class and open-ended tasks [Mvot,FrSc]
- How $\alpha$ affect label accuracy and compute overhead [Mvot,J2Nm]
- Narrative / presentation feedback
   - Dependency on concept salience [FrSc]
   - Present searched label accuracy [J2Nm]


We will address the shared questions of reviewers in general responses, and provide separate responses for the other reviewer-specific questions.

---

### Author Response · Authors · 2025-11-20
**General Response (Part 1)**

**1. Evaluated tasks not hard enough / not really superhuman [Mvot,J2Nm,fmdG]**

We’d like to clarify that our experiments already include tasks that humans cannot reliably supervise. For example, humans only achieve 60% accuracy on gender prediction; human labels are noisy on preference modeling tasks like Alpaca [1].

In particular, given reviewers’ specific interests on hard reasoning tasks, we’d like to emphasize that the Claude dataset in Sec 4.4 already covers many hard-to-supervised tasks, such as judging correctness in MATH/GPQA-level hard reasoning tasks.

We further run additional experiment results on MATH-verification and AIME-verification (constructed similarly to GSM8k-verification), which is suggested by multiple reviewers (J2Nm,fmdG) as a hard task. Our method ICM nearly matches training on gold labels and significantly improves upon zero-shot baseline.

| Method                          | MATH Accuracy | AIME Accuracy |
| ------------------------------- | ------------- | ------------- |
| Zero-shot                       | 57.5 ± 1.2    | 58.8 ± 1.5    |
| Unsupervised Elicitaiton (Ours) | 74.7 ± 0.2    | 70.1 ± 2.1    |
| Golden Supervision              | 75.5 ± 0.3    | 70.9 ± 1.1    |

**2. How was data contamination ruled out [FrSc,J2Nm]**

While we cannot directly check data contamination since we don’t have access to llama pre-training corpus, there are several pieces of evidence that make data contamination less worrying.
- As shown in figure 2, the zero-shot performance of llama base models are close to randomly guessing (e.g. 60% on TruthfulQA, 50% on Alpaca, 48% on GSM8K)
- We reformat GSM8K and TruthfulQA into classification tasks. The GSM8K solutions are also newly sampled from diverse LMs. Take GSM8K for example, while Llama 3.1 8B chat achieves an accuracy of 85.4% in the original format (i.e. directly generating solutions), it only achieves an accuracy of 72% in judging solution correctness.
- Most of our experiments are based on llama models. Recent papers ([2]) show that while Qwen models have serious data leakage issues that make even optimizing with random rewards increase their performance on math/coding benchmarks, llama models do not.
- In Sec 4.4, we confirm that the production preference dataset is not involved in the pre-training corpus of Claude models.


[1] AlpacaFarm: A Simulation Framework for Methods that Learn from Human Feedback. NeurIPS 2023.

[2] Reasoning or Memorization? Unreliable Results of Reinforcement Learning Due to Data Contamination. arXiv 2025

[3] The Unreasonable Effectiveness of Entropy Minimization in LLM Reasoning. NeurIPS 2025

---

> ### Author Response · Authors · 2025-11-20
> **General Response (Part 2)**
>
> **3. Compared with more baselines [Mvot,J2Nm,fmdG]**
>
> **Comparison with self-consistency or LM-as-judge.** In Appendix E.1, we evaluate the baseline that generates training labels using GPT-4o with self-consistency sampling. Among reasonable baselines in the category of self-consistency inference, self-training, LM-as-a-judge, and RLAIF, this is essentially the strongest baseline we can think of, because:
> - GPT-4o is significantly larger and generally more capable than our Llama base models.
> - GPT-4o went through commercial post-training, which uses production-level human supervision, and  likely overlaps significantly with our evaluation tasks on aspects like helpfulness, harmlessness, and mathematical correctness. In other words, this is an unfair baseline with potential data leakage, where the model generating the training labels have plausible access to the test data.
> - We use self-consistency sampling. Following prior work [4] [5] [6], for each example, we use GPT-4o to sample K = 10 labels and do majority-voting to decide the final label.
>
> Still, our method ICM beats training on GPT-4o generated labels on all tasks. In particular, on Alpaca, GPT-4o labels and human labels lead to similar performance through fine-tuning, potentially suggesting that commercial post-trained models’ capability in judging helpfulness and harmlessness is bottlenecked by its post-training human data.
>
>
> | Benchmark         | Method                 | Accuracy   |
> | ----------------- | ---------------------- | ---------- |
> | GSM8K             | Golden Label           | 81.4 ± 1.4 |
> |                   | GPT-4o generated label | 79.0 ± 0.7 |
> |                   | ICM (Ours)        | 81.0 ± 0.8 |
> | TruthfulQA        | Golden Label           | 92.0 ± 1.0 |
> |                   | GPT-4o generated label | 81.9 ± 1.6 |
> |                   | ICM (Ours)                   | 90.9 ± 0.6 |
> | Alpaca            | Human Label            | 65.5 ± 0.6 |
> |                   | GPT-4o generated label | 65.2 ± 0.5 |
> |                   | ICM (Ours)                   | 68.0 ± 0.7 |
> | Gender Prediction | Golden Label           | 80.5 ± 0.3 |
> |                   | GPT-4o generated label | 77.0 ± 0.0 |
> |                   | ICM (Ours)                   | 79.7 ± 0.4 |
>
> We think this result also alleviates concerns about whether our formulation of coherence is unique, compared to prior work that tries to implement self-consistency for unsupervised and weakly-supervised methods.
>
>
> **Comparison with weakly supervised baselines.** This line of work has been extensively tested in the weak-to-strong generalization paper [7]. Tricks based on weak soft labels (e.g. removing low confidence weak pseudo labels, adaptively using strong model pseudo labels) are shown to largely underperform directly training on ground truth labels, especially on fuzzy tasks like helpfulness and harmlessness reward modeling. Instead, our paper demonstrates for the first time that it’s possible to match or exceed human supervision in broad realistic settings at production scale.
>
> **Comparison with easy-to-hard generalization.** We add results of easy-to-hard generalization baselines: fine-tuning on ground truth labels on GSM8K, and evaluate the model on MATH. In particular, we study if more ground truth labels in easy domains can unlock better accuracy in hard domains. As shown in the Table, easy-to-hard generalization largely underperforms training on golden supervision in hard domains, and using more easy data does not help. In comparison, our method matches training on golden supervision.
>
> | Method                          | # Fine-tune Data | Accuracy   |
> | ------------------------------- | ---------------- | ---------- |
> | Easy-to-Hard Generalization     | 1,024 GSM8K      | 62.9 ± 1.3 |
> | Easy-to-Hard Generalization     | 4,096 GSM8K      | 60.8 ± 2.7 |
> | Easy-to-Hard Generalization     | 20,480 GSM8K     | 63.5 ± 1.7 |
> | Unsupervised Elicitation (Ours) | 1,024 MATH       | 74.7 ± 0.2 |
> | Golden Supervision              | 1,024 MATH       | 75.5 ± 0.3 |
>
> [4] Large Language Models Can Self-Improve. ICLR 2025
>
> [5] Self-Consistency Preference Optimization. ICML 2025
>
> [6] Preference optimization for reasoning with pseudo feedback. ICML 2025
>
> [7] Weak-to-Strong Generalization: Eliciting Strong Capabilities With Weak Supervision. ICML2024

---

> ### Author Response · Authors · 2025-11-20
> **General Response (Part 3)**
>
> **4. Theoretical understanding of mutual predictability [FrSc, fmdG]**
>
>
> Minimum description length (MDL) is a natural way to interpret mutual predictability. If we formalize mutual predictability in a sequential manner: $\sum_{i=0}^N \log P(y_i \mid x_i, D_{<i})$
>
> where x is the input, y is the label, D is the whole dataset, and $D_{<i}$ indicates all (x,y) pairs before $(x_i, y_i)$.
>
> This is precisely the negated MDL. Therefore, maximizing mutual predictability is minimizing MDL, i.e. minimizing the number of bits required to describe the dataset. As a result, this would regularize the model to leverage more generalizable task capabilities to compress the task dataset, thus leading to better elicitation.
>
> We will add this discussion in the camera ready version,
>
> **5. Generalization to multi-class and open-ended tasks [Mvot,FrSc]**
>
>
> Our formalization of mutual predictability is agnostic to label set size, so it’s directly applicable to multi-class labels in classification tasks. For open-ended tasks, we think the natural solution, based on how frontier LMs are trained [8][9], is to use ICM to train a better reward model, and use that reward model in RL. Sec 4.4 demonstrated this by training a Claude 4 Sonnet assistant.
>
>
> **6. Impact of $\alpha$ on label accuracy and compute [Mvot,J2Nm]**
>
> Generally speaking, a larger $\alpha$ would yield more accurate labels, but requires more forward passes on each data point, since it would induce more label rejection during label searching.
>
> Below, we report the searched label accuracy on TruthfulQA with Llama 3.1 70B, along with the number of averaged forward passes on each data point. $\alpha=1$ leads to the fastest search but lowest label quality, while $\alpha=50$ achieves a good balance.
>
> | $\alpha$                   | 1    | 10   | 50   | 500  |
> | ------------------------ | ---- | ---- | ---- | ---- |
> | Generated Label Accuracy | 78.9 | 85.9 | 89.5 | 92.1 |
> | Average # Forward Pass   | 1.05 | 1.13 | 1.66 | 2.91 |
>
> [8] Training language models to follow instructions with human feedback. arXiv 2022
>
> [9] Training a helpful and harmless assistant with reinforcement learning from human feedback. arXiv 2022

---

### Public Comment · ~Artyom_Gadetsky1 · 2025-11-23
**Missing prior art and refuting "first to show"**

This work has substantial overlap with the paper "Large (Vision) Language Models are Unsupervised In-Context Learners", published at ICLR 2025 [1].

The central premise of this submission is that the "mutual predictability score" (or joint likelihood of labels) correlates with label quality, enabling unsupervised search of high-quality labels. However, this finding is not novel and was already established in [1]. In particular, the concept of "mutual predictability" is mathematically and conceptually isomorphic to the "joint inference framework" introduced in [1]. The core observation that ground truth labels are those which maximize internal consistency across the dataset according to the pretrained model is identical. The proposed ICM method based on the simulated annealing is effectively a specific search instantiation of the proposed joint inference framework and shares algorithmic similarities with the Unsupervised ICL method (Algorithm B2 in [1]).

Furthermore, this submission claims to demonstrate for the first time that unsupervised elicitation can match or exceed human supervision. However, prior work [1] has already provided extensive empirical evidence of this phenomenon on large scale, covering:

* 16 NLP tasks, including reasoning tasks like GSM8K
* 10 Vision-Language tasks, demonstrating that joint inference works across modalities
* Studied both closed-weight models such as GPT-4 and open-weight models such as Llama or Qwen

Despite the substantial overlap, this submission fails to discuss the relationship between mutual predictability and joint inference [1].

References:

[1] Gadetsky et al. Large (Vision) Language Models are Unsupervised In-Context Learners. ICLR 2025.

---

> ### Author Response · Authors · 2025-11-25
>
> Thanks for sharing the work.
> - Conceptually, we think that mutual predictability is similar to the joint probabiilty defined in [1], with a slight difference in conditioning on all the other examples v.s. all the prevoius examples. In addition, our paper introduces logical consistency contraints, which we find is important for penalizing degenerated solutions (Figure 10).
> - Algorithmically, we think the simulated annealing approach is quite different from the simple bootstrapping approach defined in [1].
> - Empirically, we conduct fine-tuning experiments on broader tasks, from crisp reasoning tasks like MATH/AIME, to fuzzy tasks like helpfulness and harmlessness reward modeling, **even to a production-level**. The success in reward modeling task is particularlly surprising, since it is a well-known challenging benchmark for unsupervised / weakly supervised learning [2].
>
> Regarding the "first to show" claim: as clarified in Line 74-79, we explicitly acknowledge that prior unsupervised learning has achieved success in some standard academic tasks. Our work for the first time demonstrates that unsupervised learning is promising in realisitc, production-level settings. We will make this claim more clear and add these discussions with [1] in the final revision.
>
> [1] Gadetsky et al. Large (Vision) Language Models are Unsupervised In-Context Learners. ICLR 2025.
>
> [2] Weak-to-Strong Generalization: Eliciting Strong Capabilities With Weak Supervision. ICML2024

---

> > ### Author Response · Authors · 2025-11-26
> > **Replicating your paper**
> >
> > > proposed ICM method based on the simulated annealing is effectively a specific search instantiation of the proposed joint inference framework and shares algorithmic similarities with the Unsupervised ICL method
> >
> > I'm interested in the empirical efficiveness your unsupervised ICL method. I first implemented it myself but saw negative results on my datasets. So the following results are based on your open-sourced repo and the original generative GSM8K task used in your paper.
> >
> > | Model                          | Unsupervised (Your method) | Golden Supervision |
> > | ------------------------------- | ------------- | ------------- |
> > | Llama-3.1-8B                    | 52.2   | 55.3    |
> > | LLama-3.1-70B | 54.4    | 80.8    |
> > | Qwen2.5-Math-7B             |  89.8  |  89.8   |
> >
> >
> > The results indicate that the method is effective on Qwen models that are highly optimized in the math domain, but not LLama models. Qwen models have been showing weird behaviors in math/coding domains due to pre-training data contamination problems [1] [2]. This is exactly why we intend to avoid using them in our experiments.
> >
> > In addition, IIRC, your unsupervised ICL method directly operates on the test set (please correct me if I'm wrong), while our setting is different: we produce labels on the training set, without using additional computes on the test set. Otherwise, our method would outperform training on golden supervision by a large margin.
> >
> > Based on these results, I don't think this method can "match or exceed human supervision" even on simple academic benchmarks like GSM8K across different pretrained base models. I'm happy to further test your method on more realistic benchmarks like Alpaca, MATH, and AIME if anyone is interested.
> >
> >
> > [1] Spurious Rewards: Rethinking Training Signals in RLVR. arXiv 2025.
> >
> > [2] The Unreasonable Effectiveness of Entropy Minimization in LLM Reasoning. NeurIPS 2025

---

### Meta-Review · Area_Chair_2jwt · 2026-01-06

**Summary:**

Overall, this work received very mixed reviews with diverse scores, both pre- and post-rebuttal. The authors seem to generally have provided comprehensive rebuttals, which would have partially-to-fully addressed the reviewer concerns (e.g. whether tasks were "superhuman", data contamination, etc. etc.).

I also noted the authors' (private) official comment about reviewing, as well as the external public comment about prior art, and the authors' responses to that.

Overall, while the work has generally fairly solid results from an interesting idea, there are niggling (though not necessarily fatal) concerns about novelty and significance. For example, in response to reviewer/public comments, the authors wrote that "we acknowledge that ICM cannot reliably elicit arbitrary concepts" and "we explicitly acknowledge that prior unsupervised learning has achieved success in some standard academic tasks. Our work for the first time demonstrates that unsupervised learning is promising in realisitc, production-level settings". The latter quote points to scientific novelty, while the earlier quote points to significance/scope.

Ultimately, for a highly-selective venue like ICLR, both qualitatively and quantitatively (even accounting for potential score increases, noted below) this work unfortunately does not meet the high bar for acceptance, in comparison to other submissions.

**Reviewer Concerns:**

No strong remaining technical reviewer concerns per se, but pls see above regarding novelty and significance.

**Reviewer Scores:**

-- J2NM was explicit about raising their score to 8.
-- fmdG maintained their score of 2, with remaining concerns about results on AIME and Math. The authors later provided more results, but the reviewer did not (or did not have the chance to) further respond.
-- The rebuttals generally seems to addressed well the concerns of Mvot and FrSc.
-- Hence, I would hazard that there's a 50/50 chance that reviewers fmdG, Mvot and FrSc would raise their scores.

---

### Decision · Program_Chairs · 2026-01-26

Reject